# Micronutrients and the Risk of Allergic Diseases in School Children

**DOI:** 10.3390/ijerph191912187

**Published:** 2022-09-26

**Authors:** Daniela Podlecka, Joanna Jerzyńska, Khadijah Sanad, Kinga Polańska, Monika Bobrowska-Korzeniowska, Iwona Stelmach, Agnieszka Brzozowska

**Affiliations:** 1Department of Pediatrics and Allergy, Medical University of Lodz, 90-328 Lodz, Poland; 2Medical University of Lodz, 90-124 Lodz, Poland; 3Department of Environmental and Occupational Health Hazards, Nofer Institute of Occupational Medicine, 91-324 Lodz, Poland; 4Poddębice Health Care Centre, 99-200 Poddębice, Poland

**Keywords:** asthma, allergy, children, copper, selenium, vitamins, zinc

## Abstract

Microelements and vitamins are believed to have immunomodulatory effects. The aim of the study was to establish the role of antioxidants (vitamins A, E, D) and microelements such as copper (Cu), selenium (Se), and zinc (Zn) levels in allergic diseases in schoolchildren. The study uses a cohort of 80 children (40 with and 40 without allergy/asthma diagnosis) aged 9 to 12 years recruited for the Polish Mother and Child Cohort Study. At nine to twelve years old, the children were tested for microelement and vitamin content and health status (including skin-prick test and spirometry, urine cotinine level). Demographic data were collected from mothers by an allergist. The risk of asthma occurrence was found to be significantly related to the levels of Cu and Zn. The level of Cu was also particularly closely associated with allergic rhinitis and was indicated as a significant predictor of food allergy. The levels of Cu and Zn, and poor nutritional status in general, can influence the immune system and may be considered risk factors for developing asthma, allergic rhinitis and food allergy.

## 1. Introduction

Recent years have seen increasing interest in the role of nutrition in the development of allergic diseases. Essential micronutrients (microelements and vitamins), which are believed to have an immunomodulatory effect, have become a particular object of interest. An antioxidant is a substance that protects cells from the damage caused by free radicals: unstable molecules produced by oxidation during normal metabolism [1], and which can play a part in cancer, heart disease, stroke, and various aging-related diseases. The most commonly known antioxidants include beta-carotene, lycopene, vitamins A, C, E. An essential microelement is an element such as chromium, copper, cobalt, iodine, iron, selenium and zinc, which is present in the human body in very small amounts, but is important to health [2]. These microelements play a crucial role in the development of allergic diseases such as asthma and atopic dermatitis in children [1,2]. For example, vitamin E influences the development of wheezing and asthma in children [3]. Many studies have shown a relationship between serum 25-hydroxyvitamin D (25(OH)D) level, and Scoring Atopic Dermatitis (SCORAD) index or Eczema Area and Severity Index (EASI) score severity [4]. It has also been suggested that atopic children with low vitamin D serum levels have benefited from vitamin supplementation.

However, there is a lack of clear evidence in the available literature regarding the relationship between serum vitamin A and E levels and the risk of developing allergic diseases; in addition, 25-hydroxyvitamin D is inversely associated with wheezing, especially those caused by a viral infection [5]. Imbalances in vitamin and essential mineral levels can disturb immunity and may result in immunodeficiency [6], manifesting in allergic diseases such as urticaria, atopic disease, and asthma [6,7].

The results of our previous studies of the data from the Polish Mother and Child Cohort study (REPRO_PL cohort) indicate that the levels of essential microelements in the umbilical cord are related to the chance of developing allergic disease in children [8]. More specifically, higher levels of copper and zinc in the blood are associated with a greater likelihood of developing wheezing early in life [7]. In addition, a high level of copper concentration in cord blood predisposed children to allergic rhinitis and atopic dermatitis, while a low level was observed in those with asthma. Another study found low zinc concentration in umbilical cord blood in children with allergic rhinitis ages 7–9 years old [9].

The present study aims to establish the role of antioxidants such as vitamin A, D, E and essential microelements such as copper, selenium and zinc in the development of allergic diseases in children aged 9–12 years old from the previously formed REPRO_PL cohort.

## 2. Materials and Methods

### 2.1. Study Design and Participants

The Polish Mother and Child Cohort (REPRO_PL) was established in 2007 [8,9,10]. The recruitment process and follow-up procedures, together with a complete description of the methodological assumptions, have been published elsewhere [11,12]. Briefly, the mothers were recruited during the first trimester of pregnancy at maternity units or clinics in two regions of Poland (Lodz and Legnica) if they fulfilled the following inclusion criteria: single pregnancy up to 12 weeks of gestation, no assisted conception, no pregnancy complications, and no chronic diseases as specified in the study protocol [8]. Between nine and twelve years after the birth, an invitation letter was sent to mothers with an invitation to participate in follow-up examinations covering exposure (including questionnaire data) and an assessment of the child’s health status. Parents were also interviewed to collect demographic and socioeconomic data, medical and reproductive history. The current analysis is restricted to 80 children (40 with allergy/asthma diagnosis and 40 without allergy/asthma diagnosis) children between the ages of 9 and 12 years. All mothers gave their written consent to take part before the follow–up study. Approval was given by the Ethical Committee of the Nofer Institute of Occupational Medicine, Lodz, Poland (Decision No. 7/2007, 3/2008, and 22/2014), for the primary study and by the Medical Ethics Committee of the Medical University of Lodz (Decision No. RNN/388/17/KE) for the follow–up study.

### 2.2. Child Health Assessment

Health status was assessed in children between 9 and 12 years of age; briefly, a questionnaire was administered to the mothers, and this was supplemented with information from the medical chart of each child. A clinical examination was also performed by a pediatrician/allergist in the presence of the mother or a relative. This part of the questionnaire has been developed by an allergist, based on recommendations from the International Study of Asthma and Allergies in Childhood (ISAAC) and has been applied previously [13,14]. In addition, the occurrence of allergy among family members was noted. Patients were defined as having asthma, allergic rhinitis, or atopic dermatitis if they had ever been diagnosed by a physician; asthma was diagnosed based on GINA guidelines.

Patients underwent spirometry and skin prick tests (SPT), or specific IgE where it was not possible to perform the SPT. Detailed information on the methods employed is given elsewhere [15]. Urine cotinine level was also measured as a marker of tobacco smoke exposure. The following history data from the questionnaire were considered: breastfeeding, dampness, having pets at home, and present urine cotinine level.

### 2.3. Allergen Sensitization

Skin prick testing was performed using standard allergen extracts from Allergopharma (Reinbek, Germany). A reaction >3 mm in diameter above the negative control recorded at 15 min were considered positive. 

Allergen sensitization was defined as specific IgE of ≥0.35 KU/L for at least one of tested allergens (chemiluminescence method (CLIA), Immulite 2000, XPI, Siemens, Munich, Germany).

### 2.4. Lung Function, Body Plethysmography (sRtot, Rocc)

All pulmonary function tests were performed with a Master Screen unit (Erich Jaeger Gmbh-Hochberg, Friedberg, Germany), as described elsewhere, in accordance with the ATS/ERS guidelines [16].

### 2.5. Plasma Copper and Zinc Concentrations

Plasma copper and zinc concentrations were analyzed by flame atomic absorption [17]. These methods were validated using lyophilized human reference serum samples of Seronorm from Nycomed Pharma AS (Oslo, Norway) as reference material, and through participation in interlaboratory comparison trials.

### 2.6. Plasma Selenium Concentrations

Plasma Se concentration was measured by graphite furnace atomic absorption spectrometry (GFAAS) on an Unicam Solar 989 QZ (Institute for Reference Materials and Measurements attn, CRM Sales Retieseweg 111, B-2440 Geel, Belgium) apparatus according to Nève et al. [18] with modifications. The accuracy of the method was verified by using internal quality control of the certified reference material BCR-637 (IRMM, Geel, Belgium), whose reference value and measured concentration were 81.0 µg/L (in the range 74–88 µg/L) and 82.5 ± 0.7 µg/L, respectively.

### 2.7. Vitamin D, E and A Levels

Blood vitamin D, E and A levels were determined by an HPLC system integrated with UV-VIS detector range 190–800 nm [19,20].

The only vitamin D metabolite used to determine whether a patient is vitamin D deficient, sufficient or intoxicated is 25(OH)D. The obtained 25(OH)D level represents a summation of both vitamin D intake and vitamin D produced by sun exposure. In addition, 1,25-hydroxyvitamin D is the biologically active form of vitamin D. Therefore, the study examined the correlation between these two metabolites (Vit D 1,25 and Vit D25) and allergic status: all identified correlations for both metabolites are presented in the tables.

### 2.8. Cotinine Level Assessment

A urine sample was collected from all patients. A 50 mL volume of morning urine from children was collected into a 100 mL polypropylene container (Bene, Poland). All urine samples were transported to the laboratory in a cool box and stored at −20 °C until analysis. Urinary cotinine levels in ng/mL (marker of tobacco smoke exposure) were determined according to Stragierowicz et al. [21].

### 2.9. Statistical Analysis

Statistical analysis was carried out using R software, version 4.0.5. Nominal variables are presented as counts (n) with % of group, and continuous variables as arithmetic mean ± standard deviation. The distribution was determined based on the Shapiro–Wilk test and skewness and kurtosis values. The Zn, Se, Cu, vitamin D, vitamin A and vitamin E levels were compared between groups with the independent *t*-test. Their impact on the presence of allergies and allergic diseases was determined using logistic regression models. Separate models were prepared using the various allergy and allergic disease as dependent variables, with levels of Zn, Se, Cu, vitamin D, vitamin A and vitamin E as independent variables. Both unadjusted and adjusted models were created. To avoid overfitting due to too many variables, considering the sample size, only three covariates were selected for the adjusted models: mother’s age, mother’s education, income. No correction for pets or breastfeeding was needed since all patients had pets and were breastfed.

Additionally, correlation between FEV1 (Forced Vital Capacity in 1st second) and level of Zn, Se, Cu, vitamin D, vitamin A and vitamin E was verified with Pearson’s correlation coefficient. All analyses were based on significance level of 0.05.

## 3. Results

The characteristics of the study group are presented in Table 1. The mean blood concentrations of Zn, Se, Cu, vitamin D, vitamin A, and vitamin E in the case of asthma, allergic rhinitis, atopic dermatitis, food allergy, and HDM allergy are presented in Table 2.

### 3.1. Asthma

The logistic regression found that the risk of asthma was significantly related to the level of Cu, as based on an unadjusted model, OR = 0.97 CI 95% [0.95–0.99], *p* = 0.043. This relationship was not, however, confirmed after adjustment for covariates. The Zn level also impacted the occurrence of asthma, both in unadjusted and adjusted models, OR = 1.04 CI 95% [1.01–1.08], *p* = 0.034 for adjusted model, Table 3. Patients with asthma had a significantly higher level of Zn (*p* = 0.029) and a lower level of Cu (*p* = 0.039) than patients without asthma, Table 2.

### 3.2. Allergic Rhinitis (AR)

The risk of AR was significantly related to the level of Cu, as based on unadjusted model, OR = 0.97 CI 95% [0.94–0.99], *p* = 0.031. This relationship was not confirmed after considering covariates. Patients with AR had a significantly higher level of Zn (*p* = 0.045) and a lower level of Cu (*p* = 0.027) than patients without AR (Table 2).

### 3.3. Atopic Dermatitis (AD)

The logistic regression analyses indicate that AD development does not appear to be influenced by any of the tested parameters (Zn, Se, Cu, vitamin D, vitamin A and vitamin E).

### 3.4. Food Allergy

The logistic regression confirmed that Cu was a significant predictor of food allergy, both in the unadjusted and adjusted models, OR = 1.05 CI 95% [1.01–1.11], *p* = 0.020 for the latter. Patients with food allergy demonstrated a significantly higher Cu level than patients without (*p* = 0.009), Table 3.

### 3.5. Lung Function

No significant correlation was found between FEV1 and levels of Zn, Se, Cu, vitamin D, vitamin A and vitamin E (*p* > 0.05 in all cases). Table 4.

## 4. Discussion

The essential micronutrients Cu and Zn or Se have a complex influence on the immune system and it is difficult identify a specific pro-allergic mechanism. There is wide evidence that allergic disorders, such as asthma, rhinitis and atopic dermatitis, are mediated by oxidative stress. Copper is an essential transition metal that could play an important role in redox reactions, while zinc concentration has a potent immunomodulatory capacity, particularly influencing T helper cell organization and cytokine secretion. Deficiencies of the nutrients selenium (Se), zinc (Zn), and vitamins A, C, D, and E may be associated with the development of asthma and allergic disorders [9]. Our findings indicate that the risk of asthma is significantly related to the level of Cu, according to an unadjusted model, and that the level of Zn also influences the occurrence of asthma, in both unadjusted and adjusted models. What is more, the concentration of Cu was significantly related to AR and appears to be a strong predictor of food allergy. In our group no significant correlations were found between vitamin levels (A, E, D) and allergic status and spirometry parameters. 

Micronutrients are often obtained from food, and are essential for the healthy development of the body and disease prevention [22,23]. Among these, zinc, selenium, and copper are known to play important roles in maintaining health; however, all micronutrients have a significant impact on the functioning of the immune system [23,24]. Currently, supplementation of zinc or selenium in addition to food sources is recommended for well-being [25]. 

It is known that both copper deficiency and excess can lead to chronic inflammation [25,26]. Our findings indicate that Cu and Zn are both important components in the development of some allergic conditions, such as asthma, allergic rhinitis and food allergy, but not with atopic dermatitis. No differences were observed with regard to the intake of selenium or the vitamins; however, the study did not evaluate the children’s diet or supplementation by individual nutrients. Hu states that increased selenium levels are associated with a reduced prevalence of asthma and increased lung function [27]. 

Seaton et al. propose that the population has become more susceptible to asthma due to reduced antioxidant intake [28]. It has also been supposed that the lower intake of antioxidants and microelements might affect the first interaction between antigens and the immune system [29,30]. Another study found that maternal zinc intake during pregnancy was negatively associated with asthma ever diagnosed or active asthma [13], and patients with recurrent wheezing and a positive Asthma Predictive Index were found to have significantly lower vitamin D levels, and significantly higher serum zinc and copper concentrations [31]. It has also been postulated that the serum level of vitamin D, copper and zinc can be used as a routine biomarker for asthma risk in patients with recurrent wheezing alongside the Asthma Predictive Index and temporal pattern of wheeze [31].

However, these findings are not confirmed by our present results. It is possible that the increase in serum Zn in our study might cause a reduction in serum Cu, resulting in greater inflammation by decreasing the capacity of the antioxidant system. Other trials have failed to show any differences between children with asthma and healthy children in terms of Cu and Zn level [32].

The two main sources of dietary vitamin A are provitamin A carotenoids and retinol, which are mainly found in red and orange fruits [33,34]. Vitamin A is considered to be strong lipid-soluble antioxidant [33]. Gilliland et al. found a correlation between vitamin A, improved lung function and lower asthma incidence [35]. In addition, the increased risk of asthma was been found to be associated with lower serum carotenoid levels, including those of vitamin A precursors (α-carotene, β-carotene and β-cryptoxanthin), but not with the serum vitamin A concentration [36]. In a murine model of vitamin deficiency, vitamin A is linked with abnormal accumulation of airway smooth muscle and inflammation in lungs [34]. Unfortunately, human studies did not identify any relationship between serum vitamin A concentration and the development of asthma and allergic diseases in children and adults [37,38]. Furthermore, vitamin A supplementation was not found to have any significant effect in preventing early life childhood atopy up to an age of seven years in a randomized controlled trial [39]. This is consistent with our findings in a similar-aged population.

The influence of vitamin D on allergic conditions has been widely investigated in the past decade and the findings suggest great potential in achieving asthma control. In a recent meta-analysis, fourteen studies demonstrated that vitamin D supplementation for the management of asthma was associated with a lower rate of exacerbation, but that it would be helpful only in patients with bronchial asthma with vitamin D insufficiency at baseline (vitamin D insufficiency defined as a concentration in plasma less than 20 ng/mL) [40]. The same meta-analysis found FEV1% improvement in patients with air limitations to be associated with and vitamin D insufficiency [41]. Vitamin D supplementation has also been found to be associated with a lower rate of exacerbation and an improved pulmonary function in adults [42,43,44]. Similar conclusions were found also in three different pediatric studies by Kerely et al. and Majak et al. [45,46,47]. In addition, a brief review of several observational studies including both child and adult patients found that higher vitamin D levels were associated with a lower risk of acute asthma exacerbations [48,49]. 

Unfortunately, no uniform results have been obtained for vitamin D concentration and asthma control. While a positive relationship has been found between 25(OH)D level and childhood asthma control in some studies [31,50,51], no correlations between vitamin D concentrations and asthma control were noted in others [52,53,54]. While no correlations were found between allergic diseases and vitamin D level at the age of 9–12 years in the present study, low concentrations of vitamin D were found to be linked with wheezing and food allergy in the same population at the age of two years [46].

Vitamin E is a natural antioxidant found in large amounts in vegetables and fruits [55,56]. It has been found to play a crucial role in fetal lung development and vitamin E supplementation was associated with lower immunoglobulin E levels, suggesting a probable protective effect at the age of asthma onset [20]. The typical Western diet, low in natural antioxidants, has been linked with an increased incidence of asthma [56], and Misso et al. found low plasma vitamin E levels to be associated with worse lung function and increased severity of asthma symptoms [57]. In addition, Fogarty et al. indicate that vitamin E supplementation can reduce asthma severity and improve lung function [58]. A meta-analysis of observational studies found asthma to be related to vitamin A and C intake, but not vitamin E [59]. No correlations between atopy status and vitamin E were found in our previous study [59].

Our present findings do not indicate any correlations between antioxidant vitamins or micronutrients, and FEV1 and FVC. Although these findings may be due to the small sample size of asthma patients, they are consistent with those of other researchers [33,56].

Our study has some limitations. Firstly, the group sample is rather small, and secondly, the plasma concentrations of vitamins and microelements were measured at just one point of time (age 9–12 years); it would be interesting also to perform such an analysis for the whole period after birth. In addition, no data was obtained on the diet of the children, nor any micronutrient or vitamin supplementation nor the current nutritional status of the investigated patients. As only blood vitamin and micronutrient levels are included in the study, it is not possible to speculate on the reason for the low vitamin and micronutrient status of children. It seems also important to underline correlation does not necessarily imply a direct causal relationship. 

Nevertheless, the key strength of the study is its diagnosis of allergic disease relying on both clinical assessment and evidence of allergic sensitization and assessment of skin-prick tests and pulmonary function tests; this allowed an objective outcome evaluation.

## 5. Conclusions

In conclusion, low concentrations of microelements, especially those of Cu and Zn, can influence the immune system, and may be considered as risk factors for the development of asthma, allergic rhinitis and food allergy.

## Figures and Tables

**Table 1 ijerph-19-12187-t001:** Characteristics of study group.

Variable	Total Group	Asthma	Allergic Rhinitis	Atopic Dermatitis	Food Allergy	HDM Allergy
n	82	41	41	16	14	20
Sex, female, n (%)	47 (57.3)	23 (56.1)	24 (58.5)	7 (43.8)	6 (42.9)	9 (45.0)
Age of mother, years, mean ± SD	28.38 ± 4.44	27.11 ± 4.35	27.07 ± 4.70	28.45 ± 4.47	27.20 ± 3.41	26.61 ± 3.16
BMI *, mean ± SD	21.47 ± 3.82	20.56 ± 4.05	20.78 ± 4.28	19.28 ± 5.45	20.65 ± 2.26	20.30 ± 1.81
Mother’s education, n (%)						
≤9 years of education	12 (14.6)	12 (29.3)	11 (26.8)	4 (25.0)	4 (28.6)	8 (40.0)
10–12 years of education	21 (25.6)	10 (24.4)	9 (22.0)	3 (18.8)	1 (7.1)	1 (5.0)
>12 years of education	49 (59.8)	19 (46.3)	21 (51.2)	9 (56.3)	9 (64.3)	11 (55.0)
Socio-economic status, n (%)						
Modest	38 (46.3)	23 (56.1)	20 (48.8)	10 (62.5)	10 (71.4)	9 (45.0)
High	44 (53.7)	18 (43.9)	21 (51.2)	6 (37.5)	4 (28.6)	11 (55.0)
Parity, n (%)						
0	44 (53.7)	26 (63.4)	26 (63.4)	8 (50.0)	6 (42.9)	12 (60.0)
1	10 (12.2)	10 (24.4)	9 (22.0)	3 (18.8)	2 (14.3)	5 (25.0)
>1	28 (34.1)	5 (12.2)	6 (14.6)	5 (31.3)	6 (42.9)	3 (15.0)
Astma, n (%)	41 (50.0)	41 (100.0)	35 (85.4)	13 (81.3)	11 (78.6)	18 (90.0)
Allergic rhinitis, n (%)	41 (50.0)	35 (85.4)	41 (100.0)	8 (50.0)	7 (50.0)	19 (95.0)
Atopic dermatitis, n (%)	16 (19.5)	13 (31.7)	8 (19.5)	16 (100.0)	9 (64.3)	7 (35.0)
Food allergy, n (%)	14 (17.1)	11 (26.8)	7 (17.1)	9 (56.3)	14 (100.0)	7 (35.0)
HDM allergy, n (%)	20 (24.4)	18 (43.9)	19 (46.3)	7 (43.8)	7 (50.0)	20 (100.0)
Skin prick test, n (%)	43 (52.4)	36 (87.8)	41 (100.0)	9 (56.3)	7 (50.0)	20 (100.0)
Dampness, n (%)	27 (32.9)	22 (53.7)	18 (43.9)	10 (62.5)	9 (64.3)	17 (85.0)
Breastfed, n (%)	82 (100.0)	41 (100.0)	41 (100.0)	16 (100.0)	14 (100.0)	20 (100.0)
Pets, n (%)	82 (100.0)	41 (100.0)	41 (100.0)	16 (100.0)	14 (100.0)	20 (100.0)
Cotinine _child_urine_ng/mL, geometric mean ± SD	1.88 ± 3.15	2.16 ± 3.43	2.06 ± 3.73	2.13 ± 1.85	1.70 ± 1.76	2.35 ± 3.24
FEV1% (B/P ^), mean ± SD	101.93 ± 16.61	96.79 ± 17.32	96.65 ± 15.91	108.43 ± 19.01	105.96 ± 20.33	99.82 ± 15.76
FVC% (B/P ^), mean ± SD	92.46 ± 15.25	92.27 ± 15.14	91.14 ± 14.21	98.92 ± 18.30	95.38 ± 19.17	94.46 ± 14.22
Se (µg/L), mean ± SD	66.56 ± 11.77	64.61 ± 12.80	64.76 ± 12.40	66.75 ± 14.76	62.64 ± 12.63	62.60 ± 12.55
Cu (mg/L) mean ± SD	100.63 ± 17.46	96.67 ± 17.43	96.40 ± 18.80	103.97 ± 13.45	109.61 ± 12.04	96.28 ± 17.61
Zn (mg/L) mean ± SD	84.82 ± 15.63	88.57 ± 18.05	88.27 ± 14.78	83.66 ± 23.60	80.39 ± 16.55	89.46 ± 18.29
Vit D 25 (ng/mL), mean ± SD	22.43 ± 7.94	23.28 ± 8.87	23.48 ± 8.45	21.87 ± 7.00	22.78 ± 8.86	23.56 ± 7.94
Vit D 1,25 (ng/mL), mean ± SD	58.28 ± 20.05	56.68 ± 21.98	56.91 ± 18.88	60.33 ± 26.01	64.46 ± 25.54	57.37 ± 20.79
Vit A (µg/mL), mean ± SD	0.33 ± 0.11	0.31 ± 0.12	0.32 ± 0.13	0.34 ± 0.14	0.32 ± 0.08	0.33 ± 0.14
Vit E (µg/mL) mean ± SD	9.40 ± 2.50	8.87 ± 2.14	8.93 ± 2.61	9.67 ± 2.02	10.03 ± 1.89	9.27 ± 1.89

* BMI—Body mass index; HDM allergy—house dust mites’ allergy; ^ B/P—Best/Predicted value.

**Table 2 ijerph-19-12187-t002:** Comparison of blood concentrations of Zn, Se, Cu, vitamin D, vitamin A and vitamin E in patients with asthma, allergic rhinitis, atopic dermatitis, food allergy and HDM allergy.

Investigated Trait	Allergy/Disease Present			Allergy/Disease Absent			*p*
	Mean ± SD	95% CI	Range	Mean ± SD	95% CI	Range	
Asthma							
Se (µg/L)	64.61 ± 12.80	60.57–68.65	38.00–91.00	68.51 ± 10.44	65.22–71.81	53.00–94.00	0.135
Cu (mg/L)	96.67 ± 17.43	91.17–102.20	61.20–131.70	104.59 ± 16.77	99.30–109.90	65.40–133.60	0.039
Zn (mg/L)	88.57 ± 18.05	82.87–94.27	58.10–147.60	81.06 ± 11.84	77.33–84.80	62.20–106.00	0.029
Vit D 25 (ng/mL)	23.28 ± 8.87	20.48–26.07	7.80–52.80	21.59 ± 6.90	19.41–23.77	5.70–37.40	0.339
Vit D 1,25 (ng/mL)	56.68 ± 21.98	49.74–63.62	27.80–134.30	59.87 ± 18.04	54.18–65.57	31.50–114.40	0.475
Vit A (µg/mL)	0.31 ± 0.12	0.27–0.35	0.14–0.64	0.35 ± 0.10	0.32–0.38	0.19–0.68	0.146
Vit E (µg/mL)	8.87 ± 2.14	8.19–9.54	5.16–14.80	9.93 ± 2.73	9.07–10.79	4.44–19.04	0.053
Allergic rhinitis							
Se (µg/L)	64.76 ± 12.40	60.84–68.67	38.00–91.00	68.37 ± 10.97	64.90–71.83	53.00–94.00	0.167
Cu (mg/L)	96.40 ± 18.80	90.46–102.30	61.20–133.60	104.86 ± 15.08	100.10–109.60	78.20–133.50	0.027
Zn (mg/L)	88.27 ± 14.78	83.60–92.93	58.10–130.20	81.37 ± 15.88	76.35–86.38	62.20–147.60	0.045
Vit D 25 (ng/mL)	23.48 ± 8.45	20.82–26.15	7.80–52.80	21.38 ± 7.35	19.06–23.70	5.70–37.40	0.233
Vit D 1,25 (ng/mL)	56.91 ± 18.88	50.95–62.87	27.80–103.30	59.65 ± 21.30	52.93–66.37	31.50–134.30	0.539
Vit A (µg/mL)	0.32 ± 0.13	0.28–0.36	0.14–0.68	0.34 ± 0.10	0.31–0.37	0.19–0.63	0.318
Vit E (µg/mL)	8.93 ± 2.61	8.11–9.76	5.16–19.04	9.87 ± 2.31	9.14–10.60	4.44–15.01	0.090
Atopic dermatitis							
Se (µg/L)	66.75 ± 14.76	58.88–74.62	38.00–94.00	66.52 ± 11.07	63.79–69.24	40.00–91.00	0.953
Cu (mg/L)	103.97 ± 13.45	96.80–111.10	80.70–131.70	99.82 ± 18.30	95.32–104.30	61.20–133.60	0.314
Zn (mg/L)	83.66 ± 23.60	71.09–96.24	58.10–147.60	85.10 ± 13.25	81.84–88.36	63.00–130.20	0.817
Vit D 25 (ng/mL)	21.87 ± 7.00	18.14–25.60	7.80–35.80	22.57 ± 8.20	20.55–24.58	5.70–52.80	0.732
Vit D 1,25 (ng/mL)	60.33 ± 26.01	46.47–74.19	27.80–134.30	57.78 ± 18.53	53.22–62.34	29.80–114.40	0.715
Vit A (µg/mL)	0.34 ± 0.14	0.27–0.41	0.15–0.64	0.33 ± 0.11	0.30–0.35	0.14–0.68	0.722
Vit E (µg/mL)	9.67 ± 2.02	8.60–10.75	5.75–13.40	9.33 ± 2.61	8.70–9.97	4.44–19.04	0.571
Food allergy							
Se (µg/L)	62.64 ± 12.63	55.35–69.94	38.00–83.00	67.37 ± 11.52	64.58–70.16	40.00–94.00	0.213
Cu (mg/L)	109.61 ± 12.04	102.70–116.60	93.00–131.70	98.78 ± 17.89	94.45–103.10	61.20–133.60	0.009
Zn (mg/L)	80.39 ± 16.55	70.84–89.95	58.10–115.30	85.73 ± 15.41	82.00–89.46	63.00–147.60	0.281
Vit D 25 (ng/mL)	22.78 ± 8.86	17.66–27.89	7.80–37.40	22.36 ± 7.81	20.47–24.25	5.70–52.80	0.872
Vit D 1,25 (ng/mL)	64.46 ± 25.54	49.71–79.20	31.50–134.30	57.01 ± 18.70	52.48–61.53	27.80–114.40	0.316
Vit A (µg/mL)	0.32 ± 0.08	0.28–0.37	0.15–0.42	0.33 ± 0.12	0.30–0.36	0.14–0.68	0.758
Vit E (µg/mL)	10.03 ± 1.89	8.93–11.12	6.94–14.52	9.27 ± 2.60	8.64–9.90	4.44–19.04	0.217
HDM allergy							
Se (µg/L)	62.60 ± 12.55	56.73–68.47	38.00–88.00	67.84 ± 11.32	64.96–70.71	42.00–94.00	0.107
Cu (mg/L)	96.28 ± 17.61	88.04–104.50	68.00–124.00	102.03 ± 17.32	97.64–106.40	61.20–133.60	0.211
Zn (mg/L)	89.46 ± 18.29	80.90–98.02	58.10–130.20	83.32 ± 14.53	79.63–87.01	62.20–147.60	0.182
Vit D 25 (ng/mL)	23.56 ± 7.94	19.84–27.27	7.80–42.30	22.07 ± 7.97	20.04–24.09	5.70–52.80	0.473
Vit D 1,25 (ng/mL)	57.37 ± 20.79	47.63–67.10	32.40–103.30	58.57 ± 19.97	53.50–63.64	27.80–134.30	0.821
Vit A (µg/mL)	0.33 ± 0.14	0.27–0.40	0.14–0.64	0.33 ± 0.10	0.30–0.36	0.19–0.68	0.938
Vit E (µg/mL)	9.27 ± 1.89	8.39–10.16	5.35–13.40	9.44 ± 2.67	8.76–10.12	4.44–19.04	0.755

SD—standard deviation, CI—confidence interval. Groups with and without allergy/disease compared with independent *t*-test.

**Table 3 ijerph-19-12187-t003:** Logistic regression for occurrence of asthma, Allergic rhinitis, Atopic dermatitis, food allergy and HDM allergy.

Investigated Trait	Unadjusted Models			Adjusted Models *		
	OR	95% CI for OR	*p*	OR_adj_	95% CI for OR_adj_	*p* _adj_
Asthma						
Se (µg/L)	0.97	0.93–1.01	0.136	0.97	0.93–1.02	0.221
Cu (mg/L)	0.97	0.95–0.99	0.043	0.97	0.94–1.002	0.080
Zn (mg/L)	1.04	1.004–1.07	0.036	1.04	1.01–1.08	0.034
Vit D 25 (ng/mL)	1.03	0.97–1.09	0.337	1.02	0.96–1.09	0.436
Vit D 1,25 (ng/mL)	0.99	0.97–1.01	0.471	0.99	0.96–1.01	0.438
Vit A (µg/mL)	0.05	0.01–2.60	0.149	0.02	0.001–1.04	0.093
Vit E (µg/mL)	0.83	0.67–1.09	0.060	0.89	0.71–1.08	0.264
Allergic rhinitis						
Se (µg/L)	0.97	0.94–1.01	0.167	0.97	0.93–1.02	0.266
Cu (mg/L)	0.97	0.94–0.99	0.031	0.97	0.94–1.001	0.061
Zn (mg/L)	1.003	1.001–1.07	0.053	1.03	1.01–1.07	0.059
Vit D 25 (ng/mL)	1.04	0.98–1.10	0.234	1.04	0.97–1.11	0.273
Vit D 1,25 (ng/mL)	0.99	0.97–1.02	0.535	0.99	0.97–1.02	0.502
Vit A (µg/mL)	0.13	0.002–6.55	0.315	0.08	0.001–5.47	0.257
Vit E (µg/mL)	0.85	0.69–1.02	0.096	0.90	0.73–1.09	0.305
Atopic dermatitis						
Se (µg/L)	1.002	0.96–1.05	0.943	1.01	0.96–1.06	0.786
Cu (mg/L)	1.01	0.98–1.05	0.393	1.02	0.99–1.06	0.262
Zn (mg/L)	0.99	0.96–1.03	0.741	0.99	0.95–1.03	0.811
Vit D 25 (ng/mL)	0.99	0.92–1.06	0.751	0.98	0.90–1.06	0.645
Vit D 1,25 (ng/mL)	1.01	0.98–1.03	0.647	1.005	0.98–1.30	0.742
Vit A (µg/mL)	2.81	0.02–298.44	0.671	2.58	0.01–346.84	0.711
Vit E (µg/mL)	1.06	0.84–1.30	0.621	1.09	0.86–1.38	0.467
Food allergy						
Se (µg/L)	1.09	0.86–1.38	0.467	0.97	0.91–1.02	0.234
Cu (mg/L)	1.04	1.004–1.08	0.040	1.05	1.01–1.11	0.020
Zn (mg/L)	0.97	0.93–1.01	0.244	0.98	0.93–1.02	0.424
Vit D 25 (ng/mL)	1.01	0.93–1.08	0.857	0.99	0.91–1.07	0.808
Vit D 1,25 (ng/mL)	1.02	0.99–1.05	0.211	1.02	0.99–1.05	0.202
Vit A (µg/mL)	0.53	0.002–80.02	0.812	0.24	0.001–61.18	0.638
Vit E (µg/mL)	1.12	0.89–1.40	0.306	1.21	094–1.57	0.136

Separate models were prepared for each of response variables: asthma, allergic rhinitis, atopic dermatitis, food allergy and HDM allergy, *—models adjusted for: mother’s age, mother’s education, income, OR—odds ratio, CI—confidence interval.

**Table 4 ijerph-19-12187-t004:** Correlation between FEV1 and level of Zn, Se, Cu, vitamin D, vitamin A and vitamin E.

	r	*p*
Se (µg/L)	−0.09	0.397
Cu (mg/L)	−0.01	0.954
Zn (mg/L)	−0.14	0.203
Vit D 25 (ng/mL)	−0.07	0.537
Vit D 1,25 (ng/mL)	0.06	0.574
Vit A (µg/mL)	−0.09	0.410
Vit E (µg/mL)	0.02	0.837

## Data Availability

Not applicable.

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
