# Peer review of "Micronutrients and the Risk of Allergic Diseases in School Children"

_ijerph, 2022, doi:10.3390/ijerph191912187_

Round 1
Reviewer 1 Report
This nice study shows that some vital metallic elements are associated to the development of allergy (i.c. asthma). I have the following remarks:
- Correlation does not necessarily imply a direct causal relationship, and the authors should discuss this point.
- Is there a metabolic pathway to be suspected, which could explain why especially Cu is allergenic, while Zn and Se appear to be much less involved in allergy?
Author Response
Dear Editor,
Thank you for all comments and suggestions on our manuscript.
Linguistic correction was performed by a native as suggested.
Detailed response on your questions is marked below.
Q1: Correlation does not necessarily imply a direct causal relationship, and the authors should discuss this point.
Resp: Knowing the complexity of the immune and biochemical mechanisms involved in micronutrients, it is very difficult to describe their direct effect on allergic diseases. Nevertheless, many years of observations and research as well as reports from the literature give us the opportunity to generalize some dependencies. The tool used to analyze our results was based on many years of experience available literature on the topic.
We tried to include information on this point in the discussion.
Q2: Is there a metabolic pathway to be suspected, which could explain why especially Cu is allergenic, while Zn and Se appear to be much less involved in allergy?
Resp: Both Cu and Zn or Se affect the immune system in a rather complex way. It is quite difficult to assign a specific mechanism that works pro-allergy. As we know from the literature there is wide evidence that allergic disorders, such as asthma, rhinitis, and atopic dermatitis, are mediated by oxidative stress. Copper is an essential transition metal with a great ability to be involved in redox reactions. In particular, it plays a role as a cofactor for specific enzymes involved in antioxidant metabolism, namely superoxide dismutase (Cu,Zn-SOD), a key player in the defense against oxidative stress. Copper also appears to be essential for the maintenance of immune function, and increased susceptibility to infection, as well as an inadequate immune response, especially Th-2 response, with changes in several immunological markers, have been associated with Cu deficiency.
Zinc concentration has been shown to have a potent immunomodulatory capacity, particularly influencing T helper cell organisation and cytokine secretion. Zinc deficiency leads to reduced phagocytic ability of macrophages, while zinc supplementation can improve phagocytosis but the molecular mechanisms behind how zinc affects this process is not yet known.
Most of biochemical actions of Se are exerted through a group of essential proteins (selenoproteins), where it is incorporated in the form of selenocysteine. Selenoproteins are thought to be essential components of the human antioxidant defense. Insufficient levels of Se in the human body impair cellular immunity, and individuals with low plasma concentrations of Se show increased oxidative stress, with increased production of reactive oxygen species.
The role and participation in the immunological mechanisms of individual elements are an interesting and extensive topic, but due to the journal's size requirements, the above information was only mentioned.

Reviewer 2 Report
Podlecka and colleagues present results from the REPRO_PL cohort regarding the relationship between essential micronutrient status and immune-mediated illness. This is an emerging area for which the present data will be of tremendous value to the field. This reviewer thanks these authors for their work in this area. Overall, this reviewer has few issues regarding the study design and methods, but the manuscript is in great need of review by a scientific editor. I have made note of some specific issues below:
Introduction would benefit by explaining 'essential micronutrients' rather than the several lines going back and forth between several terms (i.e. micronutrient, microelement, and the misuse of the term 'element')
Line 152: The acronym 'FEV1' needs to be defined.
Line 64: Change heading to "Study design and participants" as the term population is not relevant to this study design.
Line 65 and 232: Please ensure in-text citations are in parentheses.
Lines 135-139: This information belongs in the study design and participants section.
Lines 150-151: The fact that not all participants were breastfed or exposed to pets is why the authors should have adjusted for these covariates. Either adjust the analysis or omit this line.
Table 1: This table needs to include stratified data by whichever group represents the primary analysis so that readers can see how groups differ.
Tables 1,2,3,4: The terms 'Vit 25' and 'Vit 1,25' or incorrect. Please revise to be clear that this is in reference to vitamin D.
Throughout: The data is presented as odds ratios, yet the authors have used the term 'risk' throughout the text to present the results. This is incorrect as odds ratios refer to odds not risk.
Author Response
Dear Editor
Thank you for all comments and suggestions on our manuscript.
Linguistic correction was performed by a native as suggested.
The introduction and presented results have been improved to improve clarity.
Detailed response on your questions is marked below.
Q1 Introduction would benefit by explaining 'essential micronutrients' rather than the several lines going back and forth between several terms (i.e., micronutrient, microelement, and the misuse of the term 'element')
Resp: As requested by the previous reviewer, the introduction to the study includes an explanation of the difference in terminology (microelement, micronutrient, trace element, antioxidant). However, it seems appropriate to clarify this, therefore corrections have been made
Q 2: Line 152: The acronym 'FEV1' needs to be defined.
Resp: The acronym FEV1 was expanded and added to the text (Forced Expiratory Volume in 1 Second (lung airflow measure))
Q3: Line 64: Change heading to "Study design and participants" as the term population is not relevant to this study design.
Resp: Thank you for that comment. We have corrected to "Study design and participants"
Q4. Line 65 and 232: Please ensure in-text citations are in parentheses.
Resp: The parenthesis was skipped by mistake, this was corrected.
Q 5: Lines 135-139: This information belongs in the study design and participants’ section.
Resp:
Marked information was moved to study design and participant section.
Q 6: Lines 150-151: The fact that not all participants were breastfed or exposed to pets is why the authors should have adjusted for these covariates. Either adjust the analysis or omit this line.
Resp: We believe there is a small misunderstanding here, therefore let us clarify. All participants (100%) were breastfed and all of them had pets - this is visible is table 1. Therefore those 2 variables could not have been included as covariates in the analysis.
Q 7: Table 1: This table needs to include stratified data by whichever group represents the primary analysis so that readers can see how groups differ.
Resp: We have added to table 1 data for each subgroup. Please note that since groups are not mutually exclusive (e.g. one participant may have both asthma and food allergy), they do not sum up to total group and therefore because of that we did not calculate tests comparing subgroups.
Q8: Tables 1,2,3,4: The terms 'Vit 25' and 'Vit 1,25' or incorrect. Please revise to be clear that this is in reference to vitamin D.
Rsp: As we know from the literature 25(OH)D is the only vitamin D metabolite that is used to determine whether a patient is vitamin D deficient, sufficient, or intoxicated. 25(OH)D is a summation of both vitamin D intake and vitamin D that is produced from sun exposure. 1,25 -hydroxyvitamin D is the biologically active form of vitamin D. Thus, we tried to assess the correlation between these two metabolites of vitamin D and allergic status; in all tables all correlations are presented for both vitamin D metabolites.
Q9: Throughout: The data is presented as odds ratios, yet the authors have used the term 'risk' throughout the text to present the results. This is incorrect as odds ratios refer to odds not risk.
Resp: Thank you for this comment. This has been corrected in whole manuscript.

Round 2
Reviewer 2 Report
This reviewer thanks the authors for their prompt response and apologizes for the misunderstanding regarding the breastfed and pet covariates.
All corrections are adequate with the exception of "Vit 25" and "Vit 1,25". These two terms are incomplete and must be corrected to "Vit D 25" and Vit D 1,25" as there is no such thing as vitamin 25 or vitamin 1,25. This correction must be made in all tables and throughout text.
Author Response
Dear Reviewer,
thank you for your comments. "Vit 25" and "Vit 1,25"were corrected to "Vit D 25" and Vit D 1,25 as suggested. All corrections are marked in green.